# Low-Density Neutrophils Contribute to Subclinical Inflammation in Patients with Type 2 Diabetes

**DOI:** 10.3390/ijms25031674

**Published:** 2024-01-30

**Authors:** Benjamin L. Dumont, Paul-Eduard Neagoe, Elcha Charles, Louis Villeneuve, Jean-Claude Tardif, Agnès Räkel, Michel White, Martin G. Sirois

**Affiliations:** 1Research Center, Montreal Heart Institute, Montreal, QC H1T 1C8, Canada; benjamin.dumont@umontreal.ca (B.L.D.); paul-eduard.neagoe@icm-mhi.org (P.-E.N.); cha_cha1993@hotmail.com (E.C.); louis.villeneuve@icm-mhi.org (L.V.); jean-claude.tardif@icm-mhi.org (J.-C.T.); 2Department of Pharmacology and Physiology, Faculty of Medicine, Université de Montréal, Montreal, QC H3C 3J7, Canada; 3Department of Medicine, Faculty of Medicine, Université de Montréal, Montreal, QC H3T 1J4, Canada; agnes.rakel@umontreal.ca; 4Research Center, Centre Hospitalier de l’Université de Montréal (CHUM), Montreal, QC H2X 0A9, Canada

**Keywords:** type 2 diabetes, neutrophil, cytokines, NETs

## Abstract

Type 2 diabetes (T2D) is characterized by low-grade inflammation. Low-density neutrophils (LDNs) represent normally less than 2% of total neutrophils but increase in multiple pathologies, releasing inflammatory cytokines and neutrophil extracellular traps (NETs). We assessed the count and role of high-density neutrophils (HDNs), LDNs, and NET-related activities in patients with T2D. HDNs and LDNs were purified by fluorescence-activated cell sorting (FACS) and counted by flow cytometry. Circulating inflammatory and NETs biomarkers were measured by ELISA (Enzyme Linked Immunosorbent Assay). NET formation was quantified by confocal microscopy. Neutrophil adhesion onto a human extracellular matrix (hECM) was assessed by optical microscopy. We recruited 22 healthy volunteers (HVs) and 18 patients with T2D. LDN counts in patients with diabetes were significantly higher (160%), along with circulating NETs biomarkers (citrullinated H3 histone (H3Cit), myeloperoxidase (MPO), and MPO-DNA (137%, 175%, and 69%, respectively) versus HV. Circulating interleukins (IL-6 and IL-8) and C-Reactive Protein (CRP) were significantly increased by 117%, 171%, and 79%, respectively, in patients compared to HVs. Isolated LDNs from patients expressed more H3Cit, MPO, and NETs, formed more NETs, and adhered more on hECM compared to LDNs from HVs. Patients with T2D present higher levels of circulating LDN- and NET-related biomarkers and associated pro-inflammatory activities.

## 1. Introduction

Diabetes is a chronic disease characterized by elevated blood glucose concentration related to the effects of abnormal pancreatic β-cell biology or insulin action [1,2,3]. Diabetes represents a major burden to healthcare systems [1,4,5,6]. In 2021, 537-million people worldwide were diagnosed with diabetes, resulting in health expenditures of USD 966 billion globally. Health expenditures have been forecasted to reach more than USD 1054 billion by 2045 [7]. Diabetes is also a major risk factor for ischaemic heart disease and stroke [8], which are the leading and second-leading causes, respectively, of global disease burden [9]. Subclinical chronic inflammation is associated with the development and presence of T2D [10,11,12]. Elevated levels of several inflammatory biomarkers, at baseline in different human populations, are predictive of T2D occurrence and complications [11,12]. Moreover, high levels of IL-6 and CRP are significantly associated with an increased risk of T2D [13,14,15]. A greater amount of white blood cells (WBCs) is also associated with a worsening of insulin sensitivity, predicting the development of T2D [16]. A significant association with T2D has been observed for both neutrophil and lymphocyte counts, but not for monocyte [17,18,19].

Neutrophils are first-line responders of the innate immune system. Circulating neutrophils are classified in two subsets: the high-density neutrophils (HDNs) and the low-density neutrophils (LDNs) [20,21]. LDNs normally represent less than 2% of neutrophils in healthy individuals [20,21], but their counts are increased in multiple pathological disorders [21,22]. In addition, LDNs are more potent than HDNs in enhancing inflammation by producing cytokines [22] and neutrophil extracellular traps (NETs) (NETosis) [23,24]. NETs are composed of double-stranded DNA decorated with pro-inflammatory cytokines and enzymes such as myeloperoxidase (MPO) [25]. Upon their release, NETs can bind to endothelial cells (ECs) through the von Willebrand factor (vWF) [26,27] and P-selectin [28,29], providing a scaffold for the binding of platelets, neutrophils, and erythrocytes, leading to fibrin deposition and thrombotic microvascular occlusion [24,30].

Neutrophils isolated from patients with T2D are more susceptible to NETosis since this process was shown to metabolically require glucose [24,31]. It has also been shown that NETs contribute to end-organ damage in patients with T2D; for instance, in diabetic retinopathy [32,33].

The contribution of LDNs to the aforementioned inflammatory activities have not been studied in patients with T2D. Due to the increases in LDNs and pro-inflammatory profile of LDNs in pro-inflammatory pathologies as observed in obesity, hypertension, and cancer [21,34,35], we hypothesized that LDNs could play a role in the chronic inflammation seen in T2D. Consequently, our objective was to assess if circulating LDN counts in T2D, and their biological activities, especially NETosis, are increased in these patients.

## 2. Results

### 2.1. Study Population

A total of 40 participants were enrolled, including 22 healthy volunteers (HVs) and 18 patients with Type 2 diabetes. The baseline characteristics are provided in Table 1. The age of patients with T2D was significantly higher than the HV group. However, Pearson correlation analyses showed no significant correlation between all circulating biomarkers (including LDNs counts) and age in HV, except for MPO-DNA. Most patients with T2D suffered from dyslipidemia (85%) and were taking statins (92%). There were no significant differences between HVs and patients with T2D in hematology parameters. All patients with T2D received antidiabetic medications, with metformin and an SGLT2 inhibitor being used most commonly.

### 2.2. Circulating Counts of LDNs, HDNs, and Lymphocytes

From the complete blood count (CBC) measurements (Table 1), the total counts of neutrophils and lymphocytes in HVs (4.21 × 10^9^/L and 1.62 × 10^9^/L) and patients with T2D (4.28 × 10^9^/L and 1.57 × 10^9^/L) were similar. Thus, the ratio of total neutrophils over lymphocytes (N/L) were not significantly different (2.61 for HVs vs. 2.93 for patients with T2D) (Table 1). Using flow cytometry, we assessed the total number of neutrophils (HDNs + LDNs), HDNs, and LDNs in both groups (Figure 1). Compared to HV, the total count of neutrophils and HDNs was similar in patients with T2D (Figure 1A,B). The count of LDNs was significantly higher (2.6-fold) in patients with T2D compared to HVs (*p* < 0.05) (Figure 1C). Similarly, the total neutrophils and HDNs to lymphocyte ratio were similar between HVs and patients with T2D. However, the ratio of LDNs to lymphocytes was significantly increased in patients with T2D by 2.7-fold (*p* < 0.01; Figure 1D–F).

### 2.3. Circulating Biomarkers of Inflammation and NETs

Circulating NETs were measured using H3Cit and MPO-DNA, two specific NETs components, and MPO, an enzyme involved in NET formation [28,29,36]. All three biomarkers were significantly increased in patients with T2D (H3Cit by 2.4-fold, MPO-DNA by 1.7-fold, and MPO by 2.8-fold) (Figure 2A–C). In patients with T2D, we also observed a significant increase of circulating IL-6 (2.2-fold, *p* < 0.001), IL-8 (2.7-fold, *p* < 0.001), and CRP (1.8-fold, *p* < 0.01), which are known for their capacity to induce NETosis [25,37,38] (Figure 2D–F). Due to the link between T2D and cardiovascular diseases, we assessed the NT-proBNP and troponin T levels (Figure 2G,H), markers of cardiac stress and lesion, respectively. Both biomarkers were significantly increased in patients with T2D by 2.6-fold (*p* = 0.0089) and 2.4-fold (*p* < 0.001), respectively, compared to HV. Finally, the circulating granulocyte colony stimulating factor (G-CSF) was quantified to ensure that the LDN increase observed in patients with T2D was not due to infection-associated inflammation [39,40,41]. G-CSF levels were not significantly different between HVs and patients with T2D (Figure 2I).

### 2.4. Basal NETosis in Isolated HDNs and LDNs

We assessed the percentage of HDNs and LDNs undergoing basal NETosis using the detection of H3Cit and MPO by flow cytometry (H3Cit+ and MPO+) (Figure 3A,C), and their corresponding expression levels measured by their respective mean fluorescence intensity (MFI) (Figure 3B,D). We observed in patients with T2D a higher percentage of H3Cit+ and MPO+ in both HDNs (3.8-fold and 6.1-fold; *p* < 0.001) and LDNs (2.1-fold and 5.9-fold; *p* < 0.001) compared to HVs (Figure 3A,C). The H3Cit+ MFI of both HDNs and LDNs from patients with T2D was significantly higher (31-fold and 116-fold) compared to their HV counterparts (Figure 3B). Furthermore, H3Cit+ MFI of LDNs from patients with T2D were also 5.4-fold higher than HDNs from the same patients (Figure 3B). The MPO MFI inter-group differences between HDNs and LDNs from HVs and T2D, respectively, were not statistically significant, whereas in both HVs and patients with T2D, the MPO MFI were highr in LDNs than HDNs (HVs by 3.2-fold; *p* < 0.001, and patients with T2D by 15.7-fold; *p* < 0.001).

### 2.5. In Vitro NETosis from Stimulated HDNs and LDNs

Neutrophils (HDNs and LDNs) were exposed to PBS (control vehicle) and pro-inflammatory agonists (PMA; 25 nM and CRP; 5 mg/mL) for 60 min, and NETs were quantified by confocal microscopy (representative images Figure 4A). The lowest level of NET synthesis was observed in HDNs from HVs under PBS stimulation (2% NETs area/cells area). PMA and CRP stimulation did not significantly increase NET synthesis in HDNs compared to PBS. LDNs from HVs synthesized more NETs than their HDNs counterparts, but this result was not significant (Figure 4B). As observed in HDNs, PMA and CRP did not significantly increase NET synthesis in LDNs compared to PBS. In patients with T2D, the basal value for NET synthesis in HDNs was at 7% of NETs area/cells area, and it was higher than in HVs (3.5-fold; *p* = 0.547). Treatment with PMA and CRP significantly increased NET synthesis in HDNs from patients with T2D (4.7- and 3.1-fold, respectively; *p* < 0.001 and *p* < 0.01) compared to PBS treatment. Compared to HDNs from HVs, the same PMA and CRP treatments in patients with T2D were significantly increased by 6.6- and 3.7-fold, respectively (*p* < 0.001). In LDNs from T2D, the stimulation with PMA and CRP also significantly increased NET production by 5.9- and 3.7-fold (*p* < 0.001 and *p* < 0.05), respectively. When compared to the LDNs from HV, PMA, or CRP stimulation significantly increased NET synthesis by 7.0- and 5.3-fold, respectively (*p* < 0.01 and *p* < 0.05). Finally, PMA-stimulated LDNs from patients with T2D were significantly higher than their HDN counterparts (2.3-fold; *p* < 0.05).

### 2.6. Binding of HDNs and LDNs onto Human Extracellular Matrix (hECM)

HDNs and LDNs were exposed to PBS, PMA (25 nM), or CRP (5 mg/mL), transferred onto hECM-coated plates, and incubated for 7.5 min. Under PBS stimulation, the adhesion was increased in HDNs (1.4-fold; *p* = 0.199) and significantly increased in LDNs (2.1-fold; *p* < 0.01) from patients with T2D compared to the HVs (Figure 5A). Following stimulation with PMA, the adhesion of HDNs increased significantly in both HVs (2.7-fold; *p* < 0.01) and patients with T2D (3.0-fold; *p* < 0.001) compared to PBS (Figure 5B). PMA is known to promote the activation of neutrophil β2-integrin (CD11b/CD18) complex and induce NETosis. Therefore, we assessed if a pre-treatment with a blocking goat anti-human CD18 Ab would reduce HDN and LDN adhesion onto hECM. NETs were also degraded with DNase I [28] to assess HDN and LDN cell-surface NET contributions to hECM adhesion. An anti-CD18 and DNase I combination was used to assess their dual capacity to prevent HDN and LDN adhesiveness. All treatments (anti-CD18 and DNase I, alone or combined) reduced basal adhesion of HDNs from HVs onto hECM by 34%, but they only significantly reduced anti-CD18 (p < 0.05). In patients with T2D, only anti-CD18 and its combination with DNase I were able to significantly reduce HDN adhesion by up to 46% (*p* < 0.01; Figure 5B). In HV, following PMA stimulation, the anti-CD18 pre-treatment was significantly reduced by 87% (*p* < 0.05), and the combination of anti-CD18 and DNase I completely blocked HDN adhesion compared to PBS and anti-DNase I pre-treatments alone (*p* < 0.01 and *p* < 0.05; Figure 5B). In patients with T2D, all pre-treatments significantly reduced HDNs adhesion to hECM by up to 74% (*p* < 0.001) versus PBS pre-treatment, and only the combination of anti-CD18 and DNase I significantly reduced HDN adhesion by 56% compared to DNase I (*p* < 0.05; Figure 5B). In LDNs, all treatments (anti-CD18 and DNase I, alone or combined) did not significantly reduce basal adhesion in HVs. In HVs, PMA-stimulated LDNs were significantly more adhesive (2.2-fold (*p* < 0.05) than PBS-treated cells, whereas there was no significant increase in LDNs from patients with T2D. In HV, only the combination of anti-CD18 and DNase I completely blocked PMA-induced adhesion in LDNs from patients with T2D (*p* < 0.05; Figure 5C). In patients with T2D, only the combination of anti-CD18 and DNase I pre-treatment significantly reduced basal (PBS) adhesion by 53% (*p* < 0.01) in LDNs. When treated with PMA, all treatments (anti-CD18 and DNase I, alone or combined) significantly reduced LDN adhesion to hECM compared to PMA with PBS pre-treatment by up to 100% (*p* < 0.001). Compared to DNase I alone, the addition of anti-CD18 (anti-CD18 + DNase I) further reduced significantly LDN adhesion by 44% (*p* < 0.01).

## 3. Discussion

Our study is the first to observe that circulating LDNs are significantly increased in patients with T2D compared to HVs, whereas circulating HDNs are about the same level in these two cohorts. Moreover, circulating inflammatory cytokines such as IL-6, IL-8, and CRP, as well as NETosis blood biomarkers, namely H3Cit, MPO-DNA, and MPO, were also significantly elevated in patients with T2D. In vitro, both post-isolation NET expression and post-incubation NETosis were higher in LDNs than HDNs, whereas the adhesiveness to hECM was lower in LDNs than HDNs in patients with T2D. Overall, low-grade inflammation observed in patients with T2D could be attributed in part by the concomitant increase in circulating cytokines, LDNs, and NETs.

Circulating LDNs have been shown to increase in patients with various comorbidities associated with T2D, such as obesity and hypertension [34,35]. Due to the fact that diabetes increases the likelihood of developing cardiovascular punctual events, such as myocardial infarcts and/or cardiac chronic pathologies such as coronary artery disease, myocardial infarction, or heart failure, the reported data here are in agreement with our latest study in patients that presented a significant increase in circulating LDNs in patients suffering from cardiac pathologies ([42] and submitted manuscript). Moreover, neutrophilia (increased neutrophil count) also correlates with higher mortality rates in heart failure patients [43,44,45]. Inflammatory pathologies such as cancer, asthma, and systemic lupus erythematosus were also shown to present an increase in LDN counts, and these LDNs were also activated and more inflammatory than their HDN counterparts [46,47,48,49].

We previously reported in HF patients that there was a significant increase of in vitro NETosis from HDNs after 60 min in the absence of stimuli, whereas the increase of NETosis in HDNs from patients with T2D was not significant. However, in this previous study, we did not isolate LDNs from these patients, and no discrimination was made between various HF phenotypes [38]. We also found in the latter study a significant positive correlation between circulating levels of IL-6, IL-8, CRP, and NETs in T2D and HF patients. In the present study, all the aforementioned circulating cytokines and NET components (H3Cit, MPO-DNA, and MPO) were significantly higher in patients with T2D compared to HVs. We also measured the circulating heart failure and cardiac damage markers (NT-proBNP and Troponin T, respectively) in patients with T2D and found that both markers are significantly increased versus HV, although still within the normal values.

In the present study, although both HDNs and LDNs from patients with T2D exhibit a significant increase of post-isolation and post-stimulation NETosis (Figure 3B and Figure 4B), LDNs synthesized more NETs than HDNs either with or without agonist stimulation. NET synthesis was previously associated with various thrombotic processes [30,50]. The increased LDN counts and NET synthesis combined with their capacity to obstruct small blood vessels could contribute to the development of T2D-related pathologies such as retinopathies, where NETs can create micro-aneurysms, leading to ocular capillary ruptures or nephropathies by obstructing the renal microcirculation. Consequently, they can also lead to progressive kidney failure. NETs also contribute to the peripheral arterial pathologies via the obstruction of the peripheral circulation, thus increasing the risk of lower limb ischemia, stroke, myocardial infarction, and embolism [50,51,52,53,54,55].

We reported that NETs induce adhesion of neutrophils onto hECM, which can be abrogated by DNase I treatment [28]. NETs can also modulate a rapid functional upregulation of neutrophil β2-integrin (CD11b/CD18) complex contributing to increased neutrophil adhesion onto hECM and endothelial cells [28,56]. Herein, we observed that isolated HDNs and LDNs from patients with T2D were slightly more pro-adhesive onto hECM compared to HVs. In HDNs and LDNs from healthy volunteers, the combination of blocking CD18 Ab and DNase I treatment was the most efficient to reduce neutrophil adhesion onto hECM, suggesting that these neutrophils were minimally activated. In patients with T2D, maximal neutrophil adhesion under PMA treatment occurred in HDNs. A pre-treatment, either with blocking CD18 Ab or DNase I alone, was equipotent to reduce both HDN and LDN adhesion, and their combination reduced even further PMA-induced adhesion. These data demonstrate that in patients with T2D, both neutrophil subtypes have a higher cell surface expression of NETs and activated the CD11b/CD18 complex [57].

In conclusion, we are the first study reporting that LDNs and selected pro-inflammatory biomarkers are significantly increased in patients with diabetes without clinical evidence of cardio-vascular disease compared to healthy volunteers. Moreover, our observations point towards the presence of significant low-grade inflammation in patients with T2D, namely by the increase of LDNs, circulating inflammatory cytokines and NET components, markers of cardiac damage, and in vitro NETosis. These significant changes are present despite a minimal increase in other bio markers such as IL-8, IL-6, CRP, NT-proBNP, and Troponin T compared with healthy subjects.

The limitations of this study are the low number of patients with diabetes along with the lack of stratification by the type of anti-diabetes medication that these patients were taking. Another limitation is the significant difference between the age of HVs and patients with T2D, although age variation in HVs does not correlate with LDN counts or the circulating biomarkers, except for MPO-DNA.

## 4. Materials and Methods

### 4.1. Population

This was a cross-sectional non-interventional study that included two cohorts: healthy volunteers (HVs; *n* = 22) and patients with Type 2 diabetes (T2D; *n* = 18). HVs were recruited at the Montreal Heart Institute (MHI), and patients with T2D were recruited either at the MHI affiliated Preventive medicine and physical activity Center (ÉPIC) or by Dr. Agnès Räkel at the Clinique d’Endocrinologie de Montréal. The blood collection was performed at the MHI. This study was approved by the Scientific Research Committee and the Ethics Committee of the MHI (ethics No. ICM #01-406 and No. ICM #12-1374) and conforms to the principles outlined in the Declaration of Helsinki. Donors were informed about the procedures and signed a written informed consent before participating in the study.

### 4.2. Selection Criteria of Healthy Volunteers and Patients

Healthy volunteers recruited in this study were included if they had no significant medical conditions and were not treated with anti-inflammatory or immunosuppressive drugs in the 2 weeks before blood collection. T2D donors had an HbA1c < 10% with no other cardiovascular, infectious, or inflammatory conditions. Patients with T2D were controlled by any available hypoglycemic medications and, as per guidelines, were treated with preventive hypertension medication. All donors were free from SARSCov2 infection for at least 3 months.

### 4.3. Serum and Neutrophil Collection

Venous blood from all participants was collected in serum-separating tubes (SST) (3.5-mL blood volume) and in 30 mL syringes (containing 5 mL of acid citrate dextrose for 25 mL of whole blood). The SST tubes were centrifuged to obtain serum, which was aliquoted and frozen at −80 °C. Neutrophils were isolated using the Ficoll–Paque gradient method, as previously described [28]. Upon isolation, neutrophils were resuspended in phenol-free RPMI-1640 medium (Cambrex Bio Science, Walkersville, MD, USA) supplemented with 25 mM HEPES (N-2-hydroxyethylpiperazine-N′-2-ethanesulfonic acid) (Sigma–Aldrich, Oakville, ON, Canada), 1% penicillin/streptomycin/Glutamax (VWR Intl., Montreal, QC, Canada), 1 mM CaCl_2_ (BDH Chemicals, Toronto, ON, Canada), and 5% FBS (Fetal Bovine serum; VWR Intl., Montreal, QC, Canada) (termed complete RPMI). Contamination by PMBCs was less than 0.1% as determined by morphological analysis and flow cytometry. Cell viability of neutrophils was greater than 98%, as assessed by Trypan blue dye exclusion assay.

### 4.4. HDN and LDNs Isolation by Cell Sorting

The HDN and PBMC fractions were resuspended at 10^7^ cells/mL in PBS and incubated for 20 min with a broad neutrophil marker (Alexa Fluor 647 mouse anti-human CD66b, clone G10F5; Sony Biotechnology Inc., San Jose, CA, USA) and a broad monocyte marker (Brilliant Violet 421 mouse anti-human CD14, clone M5E2; Sony Biotechnology Inc., San Jose, CA, USA). Using a cell sorter (BD FACS ARIA Fusion, BD Biosciences, Mississauga, ON, Canada), LDN (CD66b+ CD14−low) were sorted from PBMCs through an 85 µm nozzle. To ensure the same conditions between both neutrophil phenotype treatments, HDNs were sorted in the same way from the already pure HDNs pellets from blood isolation. The purified LDNs and HDNs were counted by hemacytometer and resuspended at 10^6^ cells/mL in complete RPMI.

### 4.5. Serum Biomarker Quantification

Biomarkers were quantified from serum for both HVs and patients with T2D. IL-6, IL-8, G-CSF, and MPO were quantified by Luminex Assay kits (R&D Systems, Minneapolis, MN, USA). MPO-DNA was quantified by ELISA using a monoclonal mouse anti-human MPO (Bio-Rad, Mississauga, ON, Canada) as coating Ab and the monoclonal mouse anti-human DNA (conjugated with peroxidase) from the Cell Death Detection ELISA kit (MilliporeSigma Canada Ltd., Oakville, ON, Canada) as detection Ab as previously described [58]. Circulating citrullinated histone H3 was measured using an ELISA Kit with a specific anti-H3cit antibody (Clone 11D3, Cayman Chemical, Ann Arbor, MI, USA). CRP was quantified by nephelometry and NT-proBNP, and troponin T was quantified by ECLIA electrochemiluminescence assay on a Cobas e411 (Roche Diagnostics, Rotkreuz, Switzerland) at the MHI clinical biochemistry laboratory (OptiLab, Montreal, QC, Canada).

### 4.6. Flow Cytometry

Non-sorted HDNs or PBMCs (10^6^ cells/mL) were incubated in PBS for 20 min at room temperature with various cell surface markers of neutrophils (Alexa Fluor 647 mouse anti-human CD66b, clone G10F5; Sony Biotechnology, San Jose, CA, USA), monocytes (Brilliant Violet 421 mouse anti-human CD14, clone M5E2; Sony Biotechnology, San Jose, CA, USA), NET components (rabbit anti-human citrullinated H3 (H3Cit); Abcam, Toronto, ON, Canada), and a neutrophil activation marker (PE mouse anti-human MPO, clone 5B8, BD Biosciences, San Jose, CA, USA). After H3Cit incubation, HDNs and PBMCs were further incubated for 20 min with a secondary antibody (Alexa Fluor 488 donkey anti-rabbit antibody, clone H+L; Jackson ImmunoResearch, West Grove, PA, USA). The following IgG was used as a control: mouse PE IgG (clone MOPC-21, BD Biosciences, Mississauga, ON, Canada), mouse Alexa Fluor 488 IgG (clone MOPC-21), mouse Brilliant Violet 421 IgG (clone M5E2), and mouse Alexa Fluor 647 IgG (clone MOPC-21, Sony Biotechnology, San Jose, CA, USA).

### 4.7. NETosis Assay by Confocal Microscopy

In 250 µL of complete RPMI medium, 50,000 HDNs or LDNs and agonists (PBS, PMA 25 nM, and CRP 5 mg/L) were incubated in 35 mm Petri dishes for 60 min at 37 °C with 5% CO_2_. A green nuclear fluorophore, non-permeable to live cells (SYTOX Green, 1 µM; Life Technologies, Burlington, ON, Canada), and a membrane-coloring agent (WGA, Wheat Germ Agglutinin; 1 µg/mL, ThermoFisher, Mississauga, ON, Canada) were added in 1× HBSS buffer as a dying solution. The supernatant of the Petri dishes was carefully removed and replaced by 250 µL of dye-containing solution. The images were then captured by confocal microscopy (LSM 710; Carl Zeiss, Toronto, ON, Canada) and set to acquire a mosaic of pictures (5 × 5 images) (Zen 2; Carl Zeiss, Toronto, ON, Canada) (magnification, 200×). To quantify the NET area, the number of green pixels (i.e., NETs colored by Sytox green) and red pixels (i.e., surface covered by neutrophils, colored by WGA) were measured using Image Pro Premier 9.3 Software (Media Cybernetics, Rockville, MD, USA) with a threshold to exclude background fluorescence. The results were presented as a percentage of the surface covered by NETs relative to the surface covered by cells.

### 4.8. LDN and HDN Adhesion on hECM

The adhesion assay was performed on 48-well plates coated at room temperature for 2 h with 125 µL/well hECM (human extracellular matrix, catalog no. 354237; Corning, Bedford, MA, USA) at a concentration of 20 µg/mL in RPMI, then washed with RPMI and dried for 2 h. HDNs and LDNs (10^6^ cells/mL in complete RPMI media) were pre-treated with PBS or a blocking goat polyclonal anti-human ß2 integrin/CD18 Ab (@CD18, 2 µg/mL, Gln23-Asn700, accession number AAA59490; R&D Systems, Minneapolis, MN, USA) and/or DNase I (50 U/mL; MilliporeSigma Canada Ltd., Oakville, ON, Canada) for 15 min at 37 °C.

Human ECM wells were rinsed and filled with 200 µL of complete RPMI media. Agonists (PBS, PMA 25 nM and CRP 5 mg/L) and 50 µL of HDNs and LDNs (50,000 cells) from each of the previous pre-treatments were added to the wells. After a 7.5 min incubation at 37 °C, wells were carefully rinsed with PBS to discard non-adherent neutrophils. Adhered neutrophils were then fixed with 250 µL of PBS supplemented with 2% paraformaldehyde for 20 min at room temperature, followed by an additional wash with PBS. Quantification of the adhesion was assessed using a color video digital camera adapted to a binocular microscope. For each well, four fields of view (FOV) were randomly recorded. The number of adherent neutrophils by FOV was counted using ImageJ version 1.50i (Bethesda, MD, USA), and the average number from the four FOV by well was retained [28].

### 4.9. Statistics

Group comparisons were evaluated using one-way ANOVA followed by a Dunnett post-hoc test on at least three independent experiments from independent donors. Alternatively, Brown–Forsythe correction, followed by a Dunnett T3 post-hoc, was used when applicable. T-Test was used when the analysis included only two sets of data (for HVs compared to patients with T2D for serum markers). For gender comparison, the Fisher’s exact test was used. Statistical significance was set at *p* = 0.05. Analysis was performed with GraphPad Prism 10 for Windows.

## Figures and Tables

**Figure 1 ijms-25-01674-f001:**
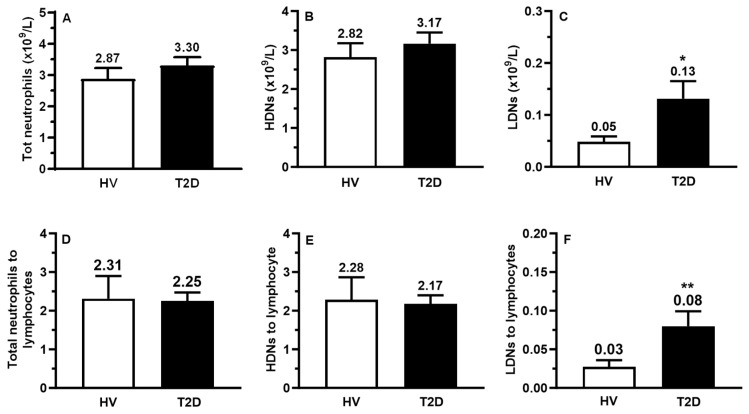
Circulating neutrophils, HDNs, LDNs, and their respective ratios to lymphocytes. Isolated (**A**) LDNs, (**B**) HDNs, and (**C**) total neutrophil counts were determined by flow cytometry, and the values obtained were divided by the HV and T2D lymphocyte counts, respectively (**D**–**F**). The data are expressed as the absolute number of cells per liter of blood (**A**–**C**) or as a ratio (**D**–**F**). All values are presented as mean ± SEM. *p* < 0.05 was considered statistically significant (* *p* < 0.05, ** *p* < 0.01 vs. HV).

**Figure 2 ijms-25-01674-f002:**
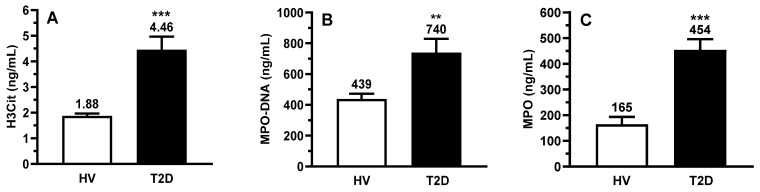
Circulating biomarkers, NETs-associated biomarkers and NETs-inducing cytokines. Circulating NETs-associated biomarkers (**A**) Citrullinated histone H3 (H3Cit), (**B**) myeloperoxidase-DNA complex (MPO-DNA) and (**C**) MPO were measured in serum by ELISA. Circulating NETs-inducing cytokines (**D**) interleukin-6 (IL-6) and (**E**) IL-8, as well as the infection marker (**I**) G-CSF, were measured by a multiplex assay, whereas (**F**) C-reactive protein (CRP) was quantified by nephelometry. NT-proBNP (**G**) and Troponin T (**H**) were measured by ECLIA (electrochemiluminescence assay). All values are presented as mean ± SEM. *p* < 0.05 was considered statistically significant (** *p* < 0.01, *** *p* < 0.001 vs. HV).

**Figure 3 ijms-25-01674-f003:**
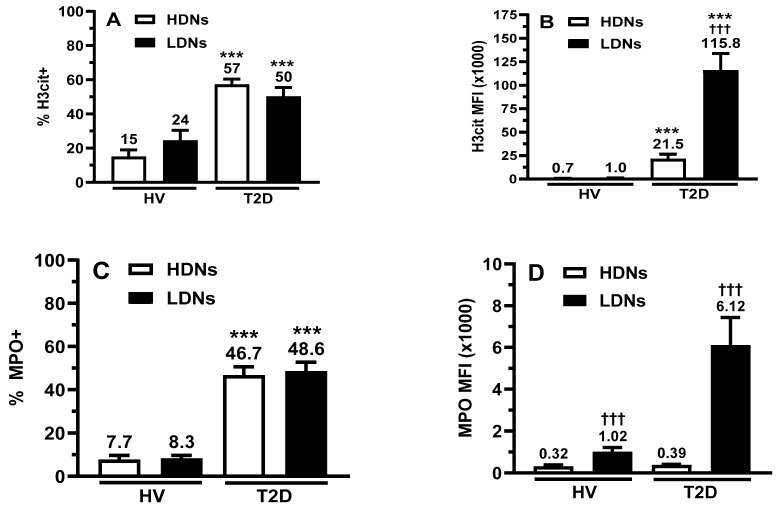
Basal NETosis in isolated HDNs and LDNs. Isolated HDNs and LDNs were incubated with two NET-associated biomarker (anti-H3Cit (**A**) and anti-MPO (**C**)), and flow cytometry was used to determine the percentage of cells in NETosis. The data are expressed as the percentage of H3Cit (**A**) and MPO (**C**) positive cells. Using the same data collected by flow cytometry, the fluorescence intensity of H3Cit (**B**) and MPO (**D**) was measured to determine the relative expression of NETs at the cell surface. The data are expressed as mean fluorescence intensity (MFI). All values are presented as mean ± SEM. *p* < 0.05 was considered statistically significant (*** *p* < 0.001 vs. HV; ††† *p* < 0.001 vs. HDNs).

**Figure 4 ijms-25-01674-f004:**
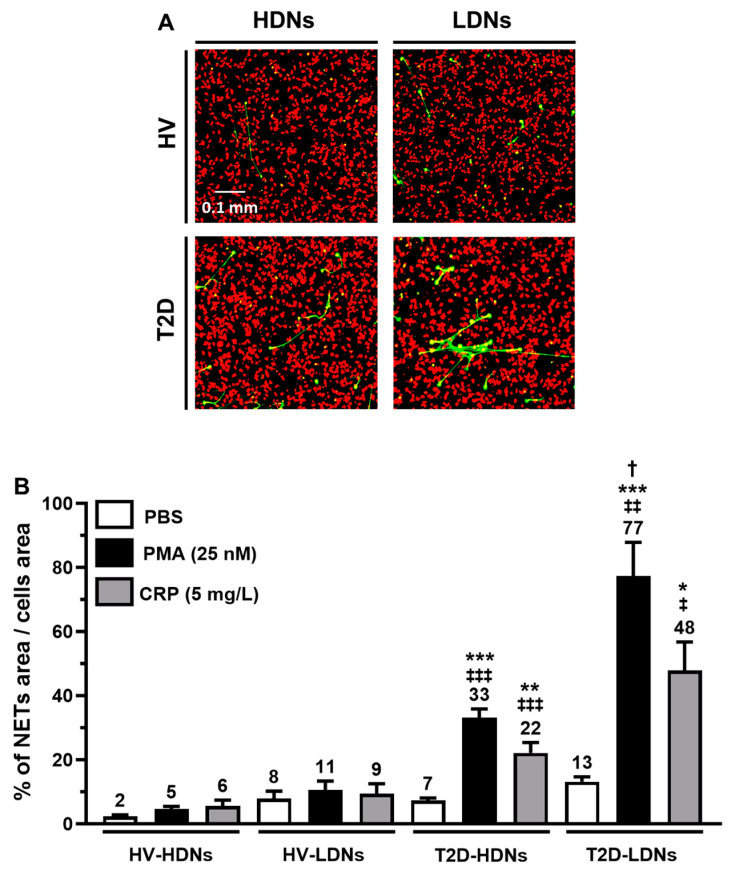
In vitro NETosis from isolated HDNs and LDNs. Isolated HDNs and LDNs were treated with PBS, PMA (25 nM), and CRP (5 mg/L) for 1 h, then stained with Sytox Green for DNA staining and WGA for cell membrane staining. The images were then captured by confocal microscopy and set to acquire a mosaic of pictures (5 × 5 images at 200× magnification). (**A**) Representative images of HDNs, LDNs (stained with WGA—red color), and NETs (stained with Sytox Green—green color). NETs were quantified by counting the number of green pixels (Sytox Green) and red pixels (WGA) using Image Pro Premier 9.3 Software with a threshold to exclude background fluorescence. (**B**) HDNs and LDNs basal (PBS) and agonist-induced NETosis were expressed as a percentage of the surface covered by NETs relative to the surface covered by cells. All values are presented as mean ± SEM. *p* < 0.05 was considered statistically significant (* *p* < 0.05, ** *p* < 0.01, and *** *p* < 0.001 vs. PBS; ‡ *p* < 0.05, ‡‡ *p* < 0.01, and ‡‡‡ *p* < 0.001 vs. HVs in corresponding treatment; † *p* < 0.05 vs. HDN corresponding agonist).

**Figure 5 ijms-25-01674-f005:**
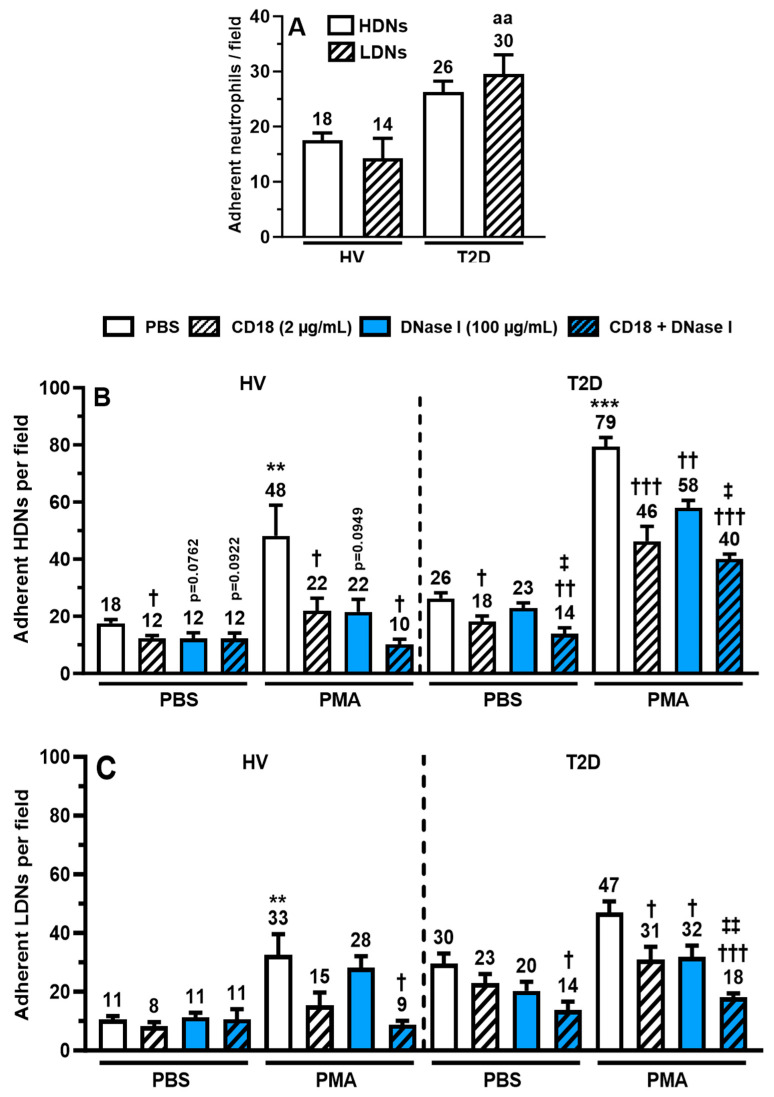
In Vitro isolated HDN and LDN adhesion to hECM. Isolated HDNs and HDNs were pre-treated with anti-human ß2 integrin/CD18 Ab and/or DNase I for 30 min prior to incubation with PBS, PMA (25 nM), and CRP (5 mg/L) for an additional 7.5 min on hECM-coated 48-plates. Adhered neutrophils were counted in four fields of view per well by optical microscopy using a digital camera. Adhesion of (**A**) basal (PBS) HDNs and LDNs from HVs and patients with T2D, agonist-induced (**B**) HDNs, and (**C**) LDNs from HVs and patients with T2D was expressed by the average number of adhered neutrophils/field from the four FOVs of each well. All values are presented as mean ± SEM. *p* < 0.05 was considered statistically significant. (aa *p* < 0.01, vs. corresponding HV; ** *p* < 0.01, *** *p* < 0.001 vs. PBS; † *p* < 0.05, †† *p* < 0.01, ††† *p* < 0.001 vs. corresponding PBS pre-treatment; ‡ *p* < 0.05 and ‡‡ *p* < 0.001 vs. DNase I).

**Table 1 ijms-25-01674-t001:** Clinical characteristics of the study population.

	HV (*n* = 22)	T2D (*n* = 18)	*p* Value
Age (years)	39.1 ± 2.9	63.4 ± 2.8	<0.001
Male	11 (50.0%)	14 (77.8%)	0.1040
Body Mass Index (BMI)	-	30.3 ± 1.3	
Smoker	-	3 (16.7%)	
Insulin treatment	-	5 (27.8%)	
Dyslipidemia	-	11 (61.1%)	
Statin	-	12 (66.7%)	
Hematology and Biochemistry			
Leukocytes (×10^9^/L)	6.18 ± 0.45	6.47 ± 0.34	0.6141
CBC neutrophils (×10^9^/L)	4.29 ± 0.29	4.28 ± 0.26	0.9833
Lymphocytes (×10^9^/L)	1.63 ± 0.04	1.57 ± 0.10	0.6046
CBC neutrophils/lymphocytes ratio	2.66 ± 0.21	2.93 ± 0.24	0.4283
HbA1c (%)	-	7.22 ± 0.28	
Glycemia (mmol/L)	-	8.67 ± 0.60	
Total cholesterol (mmol/L)	-	3.65 ± 0.13	
HDL cholesterol (mmol/L)	-	1.27 ± 0.07	
LDL cholesterol (mmol/L)	-	1.68 ± 0.12	
Triglycerides (mmol/L)	-	1.62 ± 0.21	
T2D Medication			
Metformin	-	12 (66.7%)	
Sulfonylurea	-	4 (22.2%)	
DPP-4 inhibitor	-	4 (22.2%)	
GLP-1 receptor agonist	-	7 (38.9%)	
SGLT2 inhibitor	-	9 (50.0%)	
Insulin	-	5 (27.8%)	

HV—Healthy Volunteers; T2D—Type 2 Diabetes; BMI—Body Mass Index; CBC—Complete; Blood Count; HbA1c—Glycated Hemoglobin; HDL—High-Density Lipoprotein; LDL—Low-Density Lipoprotein; DPP-4—Dipeptidyl Peptidase-4; GLP-1—Glucagon-Like; Peptide-1; SGLT2—Sodium-glucose cotransporter-2.

## Data Availability

Data are contained within the article.

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
