# Peer review of "Low-Density Neutrophils Contribute to Subclinical Inflammation in Patients with Type 2 Diabetes"

_ijms, 2024, doi:10.3390/ijms25031674_

Round 1

Reviewer 1 Report

Comments and Suggestions for Authors

Dear Authors!

    In the manuscript “Low-density neutrophils contribute to subclinical inflammation in patients with type 2 diabetes” (authors Benjamin L. Dumont et al.) , the authors present the results of the study of  interesting scientific problem namely the role of  LDNs in the chronic inflammation seen in T2D.

 This  work is scientifically useful. This research may be recommended for publication in   Int. J. Mol. Sci.  .

       However I have some comments.

1.      Table 1. Control age 39 years, diabetes - 64 years. But it would be better to study as a control people also about 64 years old without diabetes. Perhaps control at age 64 years  will be different itself from control at age  39 years. Please discuss this problem

2.      Please discuss other diseases in the manuscript, that cause similar effects . So patients with cancer had higher percentages of activated LDNs compared to healthy controls (Mauracher LM, Hell L, Moik F, Krall M, Englisch C, Roiß J, Grilz E, Hofbauer TM, Brostjan C, Knapp S, Ay C, Pabinger I. Neutrophils in lung cancer patients: Activation potential and neutrophil extracellular trap formation. Res Pract Thromb Haemost. 2023 Mar 15;7(2):100126. doi: 10.1016/j.rpth.2023.100126. PMID: 37063752; PMCID: PMC10099311).

3.      Line 21 . Abbreviation entered without full description.

4.      Line 172. …Isolated HDNs and HDNs were treated…. Are the abbreviations the same?

 The manuscript can be recommended for publication after making appropriate corrections.

 Yours sincerely,

Reviewer 2 Report

Comments and Suggestions for Authors

This is an intresting study attempting to link low-density neutrophils with the inflammatory phenomena in type 2 diabetes mellitus (DM). Overall, it's a comprehensive piece of research, but there are a few minor areas that need revision:

1.The inflammatory response is related to many serum markers, such as C-reactive protein (CRP) and procalcitonin, etc. Why have these results not been presented by the author?

2.There is a significant age difference between the two groups. How does the author address the potential reduction in research credibility due to this age discrepancy?

3.The degree of obesity is a critical data point in systemic inflammation studies. The author should not merely categorize participants as 'obese' or 'not obese' but should honestly present the average Body Mass Index (BMI) for both groups.

4.Figure 4 lacks a scale; please add one.

5.Is this a cross-sectional study? If not, should the duration of the illness and the treatments received by the patients also be indicated?
